# Genome-wide polygenic risk impact on intracranial aneurysms and acute ischemic stroke

**Eun Pyo Hong**[1], **Dong Hyuk Youn**[1], **Bong Jun Kim**[1], **Jae Jun Lee**[1], **Doyoung Na**[2], **Jun Hyong Ahn**[3], **Jeong Jin Park**[4], **Jong Kook Rhim**[5], **Heung Cheol Kim**[6], **Hong Jun Jeon**[2], **Gyojun Hwang**[7], **Jin Pyeong Jeon**[2]\*, **on the behalf of The First Korean Stroke Genetics Association Research**

1 Institute of New Frontier Research, Hallym University College of Medicine, Chuncheon, Gangwon-do, Republic of Korea, 2 Department of Neurosurgery, Hallym University College of Medicine, Chuncheon, Gangwon-do, Republic of Korea, 3 Department of Neurosurgery, Kangwon National University Hospital, Chuncheon, Gangwon-do, Republic of Korea, 4 Department of Neurology, Konkuk University Medical Center, Seoul, Republic of Korea, 5 Department of Neurosurgery, Jeju National University College of Medicine and Graduate School of Medicine, Jeju, Republic of Korea, 6 Department of Radiology, Hallym University College of Medicine, Chuncheon, Gangwon-do, Republic of Korea, 7 Department of Neurosurgery, DMC Bundang Jesaeng Hospital, Seongnam, Gyeonggi-do, Republic of Korea

\* jjpgnr@naver.com

**Data Availability Statement:** All relevant data including summary statistics are within the paper and its Supporting Information files. The original raw genotype data contains potentially sensitive data including patients' genomic profiles and its

## Abstract

Polygenic risk scores (PRSs) have an important relevance to approaches for clinical usage in intracranial aneurysm (IA) patients. Hence, we aimed to develop IA-predicting PRS models including the genetic basis shared with acute ischemic stroke (AIS) in Korean populations. We applied a weighted PRS (wPRS) model based on a previous genome-wide association study (GWAS) of 250 IA patients in a hospital-based multicenter cohort, 222 AIS patients in a validation study, and 296 shared controls. Risk predictability was analyzed by the area under the receiver operating characteristic curve (AUROC). The best-fitting risk models based on wPRSs were stratified into tertiles representing the lowest, middle, and highest risk groups. The weighted PRS, which included 29 GWASs ($p < 5×10^{-8}$) and two reported genetic variants ($p < 0.01$), showed a high predictability in IA patients (AUROC = 0.949, 95% CI: 0.933–0.966). This wPRS was significantly validated in AIS patients (AUROC = 0.842, 95% CI: 0.808–0.876; $p < 0.001$). Two-stage risk models stratified into tertiles showed an increased risk for IA (OR = 691.25, 95% CI: 241.77–1976.35; $p = 3.1×10^{-34}$; sensitivity/specificity = 0.728/0.963), which was replicated in AIS development (OR = 39.76, 95% CI: 16.91–93.49; $p = 3.1×10^{-17}$; sensitivity/specificity = 0.284/0.963). A higher wPRS for IA may be associated with an increased risk of AIS in the Korean population. These findings suggest that IA and AIS may have a shared genetic architecture and should be studied further to generate a precision medicine model for use in personalized diagnosis and treatment.

access is restricted by the Ethics Committee and Institutional Review Board (IRB) of Hallym University Chuncheon Sacred Hospital (https://chuncheon.hallym.or.kr/irb/index.asp). Interested researchers can submit "The Agreement Form of Controlled Data Usage" to the Ethics Committee and IRB to request access to the data." E-mail address for a non-author point of contact at the Ethics Committee and IRB: chuncheonirb@hallym.or.kr.

**Funding:** This study received funding from the National Research Foundation of Korea funded by the Ministry of Education (2020R1I1A3070726) (http://english.moe.go.kr/main.do?s=english). The funders had no role in study design, data collection and analysis, decision to publish, or preparation of the manuscript.

**Competing interests:** The authors have declared that no competing interests exist.

## Introduction

Several cerebrovascular diseases (CVDs) may be unexpectedly observed simultaneously in clinical practice. The prevalence of an intracranial aneurysm (IA) has been reported to be 2.0% for adults without specific risk factors but increased in patients with atherosclerosis [relative risk (RR) = 2.3, 95% confidence interval (CI): 1.7–3.1] or autosomal dominant polycystic kidney disease (RR = 4.4 95% CI: 2.7–7.2). Hurford et al. reported that the pooled mean IA prevalence in stroke or transient ischemic attack (TIA) patients was 5.1% [1]. In particular, in patients with symptomatic internal carotid artery (ICA) stenosis over 50%, the rate of incidental IA was reported to be about 4.1% [2]. These associations may be explained by the fact that IA and stroke share common genetic variants as well as clinical risk factors such as high blood pressure, high lipid levels, diabetes mellitus, and smoking [3]. However, to the best of our knowledge, there has been no genetic study on the association between IA and stroke, particularly acute ischemic stroke (AIS).

To date, robust statistical methods based on risk scoring models have been applied to the genetic architecture of complex traits. Those approaches can reduce the false-positive associations and increase the statistical power by using a large number of genetic markers in phenotypes in single nucleotide polymorphism (SNP) chip-based genome-wide association studies (GWASs) [4]. Especially, the polygenic risk score (PRS)-based model improves the identification of individual genetic risks for complex human diseases by estimating the integrative effects of multiple susceptibility loci and polygenic functional enrichment [5]. Wassertheil-Smoller et al. reported that a high PRS for depression was associated with a 3% increased chance of AIS [6]. Accordingly, the potential clinical utility of this predictive PRS may increase the understanding of the underlying mechanisms of similar or somewhat similar diseases from a genetic point of view. We previously reported the first results of a GWAS of IA in a Korean population [7] and an additional meta-analysis using two reported SNPs [8,9]. The results showed that the *GBA*, *TCF24*, *OLFML2A*, and *ARHGAP32* genes in the GWAS and the *BOLL* and *EDNRA* genes in the meta-analysis were associated with the development of IA. Based on these results, we applied PRSs obtained from IA to test whether the genetic risk factors for IA were associated with the development of AIS in a Korean population.

## Materials and methods

### Study populations

The "The First Korean Stroke Genetics Association Research" study was based on data from five university hospitals that prospectively collected the data of patients with various CVDs from March 2015 to December 2020 [7,10]. This joint study performed various genomic analyses such as GWAS, whole-exome/transcriptome sequencing, and epigenomic screening related to CVDs. The hospital-based observational cohort study enrolled 1) adult patients 18 years of age and older 2) with CVDs such as AIS, hemorrhage stroke, IA, moyamoya disease, and vascular malformations. This study also pooled control subjects who did not have CVDs or neurodegenerative diseases including dementia and Parkinson's disease. Clinical characteristics, medical information, and radiological data were collected and updated. The study including written informed patient consent was approved by the Institutional Review Board and Ethics Committee of the Hallym University Chuncheon Sacred Heart Hospital (No. 2016–3 and 2019-06-006). Detailed information including the study protocol and design has been described elsewhere [7].

### Genotyping and quality controls

Genomic DNA derived from the peripheral blood of patients was genotyped using the Axiom[TH] Asian Precision Medicine Research Array (APMRA) (Thermo Fisher Scientific, MA, USA). Among the 768 patients, the AMPRA microarray dataset of 250 IA patients and 296 controls was reported as the first IA GWAS results in 2019 [7]. High-quality plates had a plate pass-rate of $> 95\%$ for the samples and an average call rate of the passed samples of $> 99\%$. Out of 798,148 SNPs, 512,575 SNPs passed the quality control filters (i.e., a genotyping call rate of $\geq 95\%$, a minor allele frequency of $\geq 1\%$, and Hardy-Weinberg equilibrium $P$-value of $\geq 1\times10^{-6}$) [7].

### Statistical analysis

The PRS was weighted by the risk alleles and effect sizes of a total of 31 SNPs including 29 genome-wide signals ($p < 5.0\times10^{-8}$) and two reported SNPs ($p < 0.01$), which were identified in the previous studies (S1 Table) [7–9]. A total of 31 selected SNPs showed a linkage disequilibrium (LD) of less than 0.8 and were natural log-transformed by ln(odds ratio). The individual risk scores were calculated by the PLINK v1.9 score option adding a genotype imputation method under the additive inherited model (i.e., 0, 1, or 2 copies of the risk alleles) and multiplying by the effect sizes of the variants. The predictive model was calculated based on the formula shown in Fig 1. Risk assessment of the individual weighted PRSs (wPRSs) was evaluated under the generalized linear model after adjustments for age, gender, hypertension, diabetes mellitus, hyperlipidemia, smoking status, and 4 principal component analysis (PCA) values in the two-stage dataset of 250 IA and 222 AIS patients. The non-CVD control group of 296 subjects was shared by both datasets. The PCA values were estimated in the integrated dataset, which included 768 Koreans and 2,504 samples of a 1000-genome reference panel (Phase 3, version 5) (ftp://ftp.1000genomes.ebi.ac.uk/vol1/ftp/release), and shared SNPs with minor allele frequencies (MAFs) of greater than or equal to 5%. The predictability, sensitivity, and specificity were evaluated by performing area under the receiver operating characteristic curve (AUROC) analysis. These risk models have been cross validated. The best-fitting risk models based on the PRSs were stratified into tertiles of the lowest, middle, and highest risk groups. Logistic regression analysis was conducted to estimate the odds ratios (ORs) and 95% confidence intervals (CIs) after adjusting for 10 confounding factors. All univariate and multivariate analyses were performed using the STATA software package v.17.0 (Stata Corp., TX, USA).

## Results

### Baseline characteristics and polygenic risk scores

The first study stage of 250 IA patients included 104 males and 146 females with an average age of 59.3 ± 12.9 (mean ± standard deviation [SD]) years. The mean age of the 222 AIS

$$Predictor\ model\ (based\ on\ additive\ effect)$$

$$= Individual's\ polygenic\ effect\ score\ = (\sum_{i=1}^{n} wVar_i)/n$$

$i$ = Variant, $w$ = weighted (beta × -log10(p-value) × 100),
$Var$ = genotype of each variant (0, 1, or 2), $n$ = the number of variants

**Fig 1. Equation for calculating individual polygenic risk scores under the additive predictor model.**

patients in the second stage was 69.5 ± 12.9 years (118 males and 104 females). The shared controls were 52.1 ± 16.6 years old (142 males and 154 females). Detailed information on the study subjects was described elsewhere [7]. The number of summed risk alleles constructed by 31 IA-predicting variants showed a distribution of 10 to 48 and 16 to 43 copies in patients with IA and AIS, respectively (Fig 2A and 2B). The distribution of risk alleles was 10 to 43 in the 296 shared controls. The mean of these risk alleles showed approximately 37 copies in both of the two patient groups compared to 26 copies in the control group.

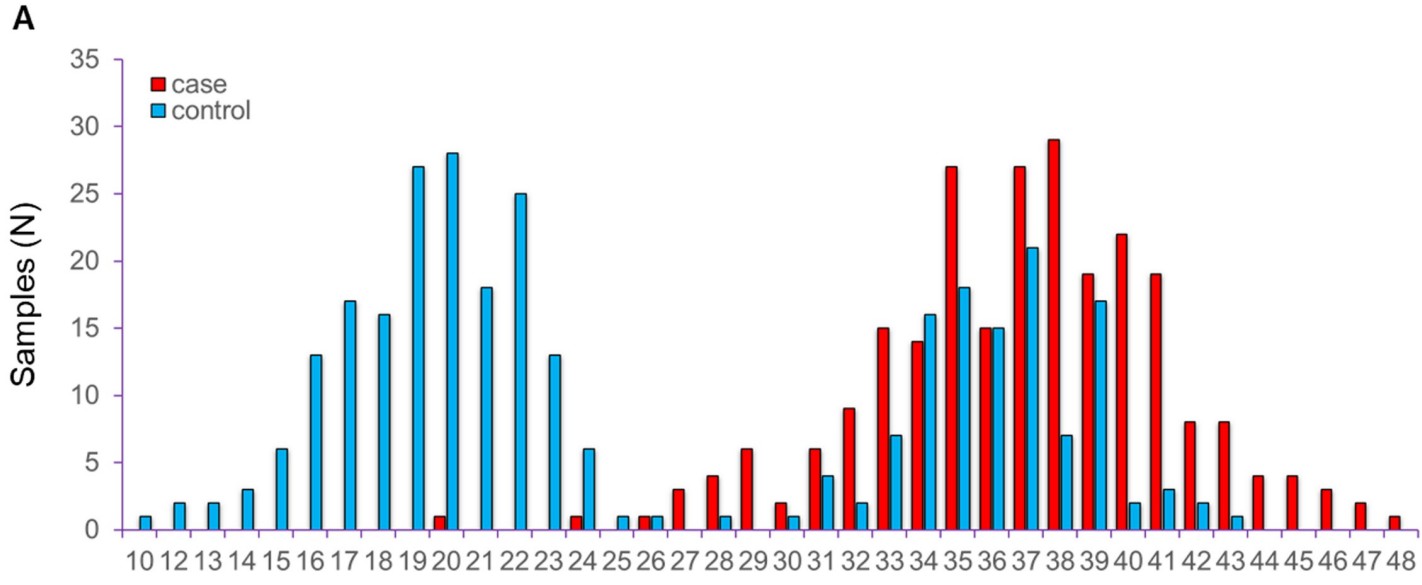

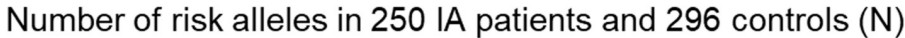

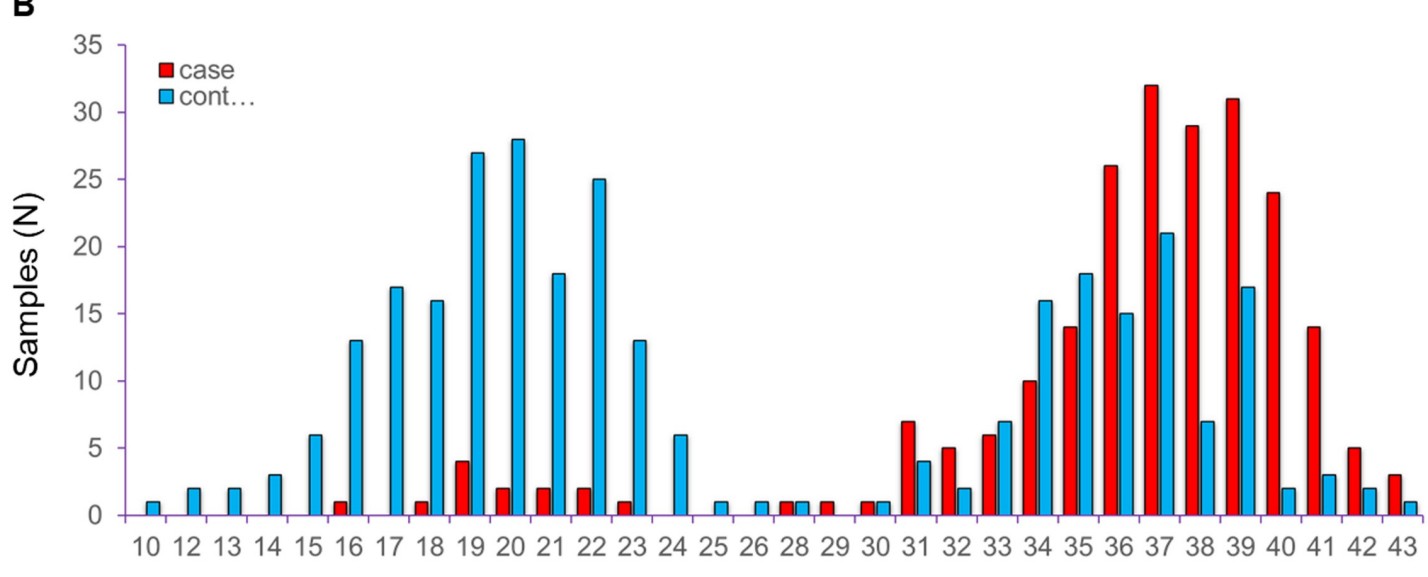

**Fig 2.** The number of summed risk alleles in (A) 250 intracranial aneurysm (IA) and (B) 222 acute ischemic stroke (AIS) patients compared to 296 shared controls. The X-axis indicates the number of risk alleles. The Y-axis indicates the sample size. The red and blue bars indicate cases and controls, respectively.

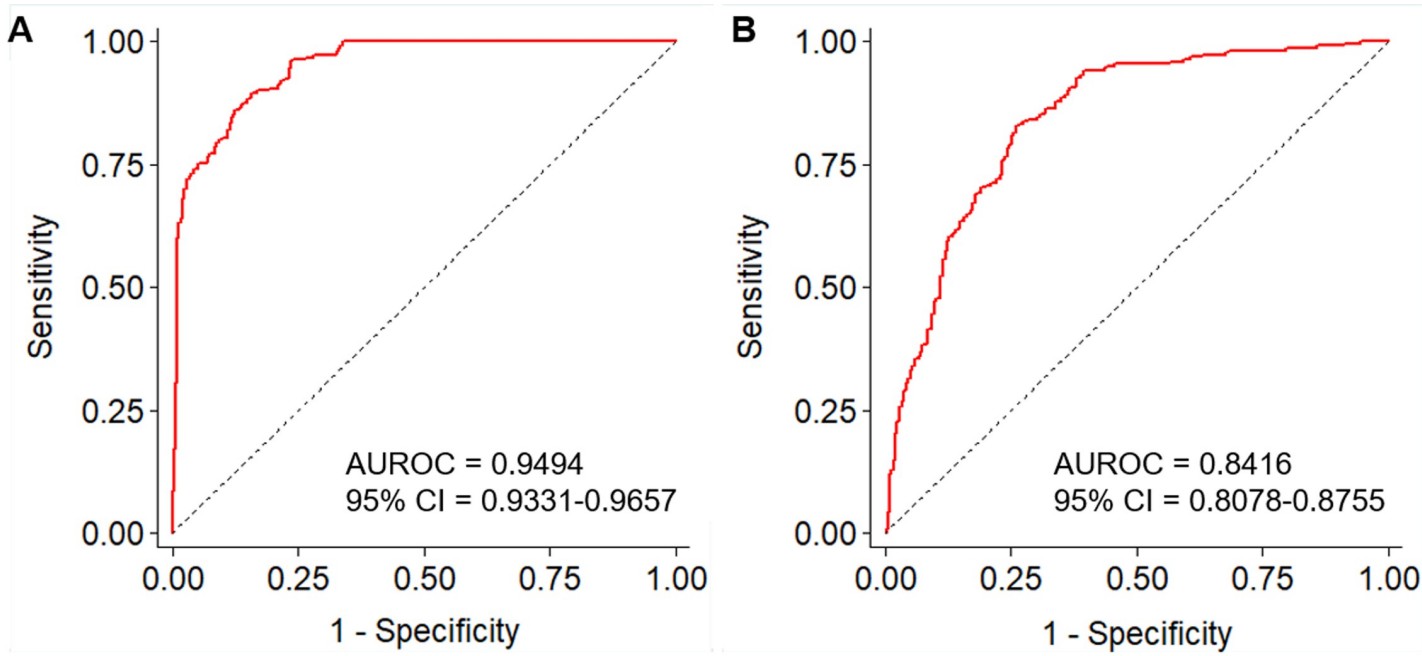

**Fig 3.** Predictability of polygenic risk scores in (A) 250 intracranial aneurysm (IA) and (B) 222 acute ischemic stroke (AIS) patients compared to 296 shared controls. The area under the receiver operating characteristic curves (AUROC) showed 0.949 for the risk of IA and 0.842 for the AIS.

### Weighted polygenic risk score in IA and stroke

The weighted PRSs (wPRSs) based on an allele scoring system and weighted by the effect size showed high predictability for IA (AUROC = 0.949, 95% CI: 0.933–0.966) (Fig 3A). This score was significantly validated to predict an increased risk of AIS (AUROC = 0.842, 95% CI: 0.808–0.876) (Fig 3B). When we stratified the individual risks of developing IA and ASI into three risk tertiles according to each of the PRS models, the risk in the T3 group was remarkably higher than that of the T1 group (OR = 691.25 and 39.76 for $PRS_{IA}$ and $PRS_{AIS}$, respectively) (Fig 4 and Table 1). Particularly, $PRS_{IA}$ still showed high clinical sensitivity, specificity, and predictability after stratification (0.728, 0.963, and 0.930, respectively). Our wPRS system contributed not only to the correct classifications of IA patients and controls (accuracy = 85.5%) but also showed a slight improvement in identifying the difference between AIS patients and controls (accuracy = 67.2%).

### Discussion

We often observe AIS patients with IA in clinical practice. Previously, in this case, the main concern was whether the acute stroke was caused by thromboembolism within an aneurysmal sac or what would happen to the IA when the patients underwent intra-arterial thrombolysis or anti-platelet agents were administered. Therefore, the main focus of this study was to identify the risk factors associated with stroke and IA. However, whether the association between these conditions is due to shared genetics has been unclear. Our study revealed that the wPRSs derived from IA patients improved the prediction of AIS, suggesting that there are shared genetic variants in the development of these two diseases.

Population-based studies have developed genetic risk models containing GWAS-driven loci, which showed a strong statistical power, to identify individuals at high risk for developing cardiovascular, cerebral, and metabolic diseases [11]. Thus, the models could be robust even

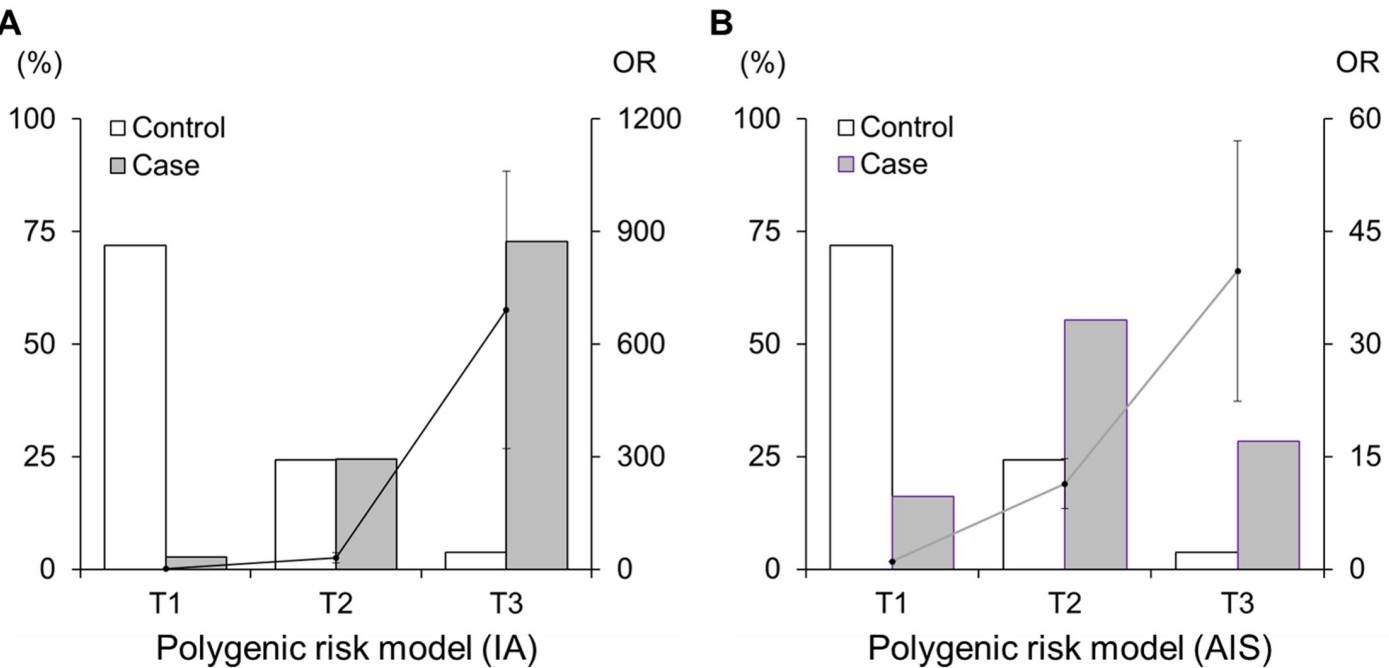

**Fig 4. The frequencies of cases and controls and their risks of diseases according to risk tertiles.** The polygenic risk model of (A) 250 intracranial aneurysm (IA) and (B) acute ischemic stroke (AIS) patients compared to 296 shared controls. The polygenic risk model was comprised of 29 genome-wide signals plus two reported loci. The white and gray bars denote the percentages of controls and cases, respectively (left Y-axis). The solid lines and dots denote the odds ratios (OR) with standard errors for each risk group compared to the lowest risk group (right Y-axis). The X-axis displays the risk tertiles (T1: tertile 1, lowest risk group; T2: tertile 2, middle risk-group; T3: tertile 3, highest risk group).

though individual variants showed a small or moderate effect size on phenotypic variance. Nevertheless, those GWA loci had insufficient predictive accuracy for the development of CVDs with complex traits, in particular, when based on the multi-locus genetic risk score (GRS) [4,12]. Hachiya et al. reported that an additional PRS model derived from one cohort

**Table 1. Weighted polygenic risk model for intracranial aneurysm (IA) and acute ischemic stroke (AIS).**

| Model[a] | Case, N (%) | Control, N (%) | OR (95% CI)[b] | $P$[b] | Sens.[c] | Spec.[c] | AUROC[c] |
|---|---|---|---|---|---|---|---|
| *IA* | N = 250 | N = 296 | | | | | |
| T1: 0.290–0.712 | 7 (2.8) | 213 (72.0) | 1.00 | | | | |
| T2: 0.712–0.789 | 61 (24.4) | 72 (24.3) | 29.84 (12.35–72.07) | $4.4×10^{-14}$ | 0.972 | 0.720 | |
| T3: 0.789–1.126 | 182 (72.8) | 11 (3.7) | 691.25 (241.77–1976.35) | $3.1×10^{-34}$ | 0.728 | 0.963 | 0.930 |
| *AIS* | N = 222 | | | | | | |
| T1: 0.290–0.712 | 36 (16.2) | | 1.00 | | | | |
| T2: 0.712–0.789 | 123 (55.4) | | 11.39 (6.43–20.17) | $7.7×10^{-17}$ | 0.838 | 0.720 | |
| T3: 0.789–1.126 | 63 (28.4) | | 39.76 (16.91–93.46) | $3.1×10^{-17}$ | 0.284 | 0.963 | 0.803 |

AUROC, area under the receiver operating characteristic curve; CI, confidence interval; OR, odds ratio; Sens., sensitivity; Spec., specificity.

[a] Weighted polygenic risk model tertile stratified into lowest risk, middle risk, and highest risk in 768 subjects containing 250 IA patients, 222 AIS patients, and 296 shared controls.

[b] OR, 95% CI, and p-value were estimated by multivariate logistic regression analysis after adjusting for age, gender, hypertension, diabetes, hyperlipidemia, smoking status, and 4 principal component values.

[c] Sensitivity, specificity, and AUROC were estimated by "*roctab*" package of STATA software.

exhibited a significant improvement in the prediction of ischemic stroke compared to multi-locus GRSs based on GWAS statistics [13]. Through follow-up research, they revealed that the genome-wide PRS was a risk factor for ischemic stroke in the general Japanese population, irrespective of environmental risk factors. Patients with the highest quintile PRSs were 2.44-fold (95% CI: 1.16–5.11) more likely to develop ischemic stroke compared to those with the lowest quintile PRSs [14]. Li et al. also reported that PRSs from the MEGASTROKE consortium were well correlated with ischemic stroke and its subtype [15]. In this study, we identified shared genetic variants for IA and AIS, which were associated with CVDs but were somewhat different in terms of pathogenesis. The wPRS model generated from IA patients was cross-validated in predicting patients with AIS, suggesting an underlying genetic link between IA and AIS in the Korean population.

Research on patient individualization, also known as precision medicine, has recently attracted much attention, although evidence-based treatment remains important. Predicting individual disease risk to prevent disease progression in susceptible individuals and early intervention and lifestyle management are at the core of precision medicine. The GRS models, which combine a small number of susceptibility SNPs identified by GWASs, have been replaced by PRS models, which were constructed to improve the statistical power by incorporating a larger number of susceptible loci that pass less stringent associations with $P$-value thresholds [16]. Accordingly, the PRS could improve the accuracy of risk stratification beyond the traditional risk factors by including variants with even modest effects on phenotypic variance. Abraham et al. reported that risk prediction based on a large number of SNPs enhanced the diagnostic accuracy of coronary heart disease prediction [17].

Ethical differences should also be considered when constructing PRS models in personalized risk prediction. Recently, the issue of the lack of generalizability of polygenic models derived from European-ancestry GWASs to other non-European populations has been raised. Thus, the importance of developing a model that considers ethnic differences has grown [18]. To use the PRS more accurately in clinical practice as precision medicine across East Asia, it is necessary to develop a PRS model based on large genomic datasets in East Asians. Additionally, risk models considering polygenic inheritance and evaluating the interactions between genetic variants and the function of metabolic, inflammatory, and immune processes in the pathogenesis of CVD may better explain an individual's risk of CVD as well as the risk of metabolic disease and coronary heart disease at the same time.

Our study had some limitations. First, the sample size was underpowered to detect polygenic variations and identify their mechanisms in the risk assessment, unlike previous Japanese and Chinese studies [13,15]. Due to the nature of bioinformatics, if the sample size of the main data in a GWAS is insufficient, the study outcomes will be underpowered. Realistically, a way to address this issue is to reduce the false positives associated with diseases by adding a fine-mapping analysis. However, fine-mapping analysis is an alternative approach to discover causal candidate variants associated with complex traits based on GWAS summary statistics [19]. Thus, the best solution would be to increase the number of patients with IA in a future study. Second, we tested the PRS in patients with AIS retrospectively based on a specific time point, although we prospectively collected the data. Third, the analysis of the AIS subtypes by PRS was not sufficiently completed. Overall, IA-predicting wPRSs increased the risk of all subtypes of AIS with predictabilities between 0.794 and 0.836 (S2 Table). Nevertheless, the sample size in each stroke subtype was small. Thus, additional analysis of a larger number of AIS patients is necessary. Accordingly, it is important to prospectively confirm the PRS model based on large-scale genomic information and validate the effects of the summed risk alleles in an independent cohort considering stroke subtypes. Despite these shortcomings, this was the

first report on polygenetic architecture and risk between IA and AIS and it should be acknowledged.

## Conclusions

We demonstrated that the PRSs obtained from IA patients can help in predicting the risk of AIS in the Korean population. Our findings showed important insights into the integrative risk by multiple susceptibility IA-predicting genes in Korean populations. The risk predictions based on the results of PRS models suggested crucial perceptions regarding ongoing attempts to identify individuals susceptible to CVD and its complications in general populations. Firstly, risk models, considering polygenic inheritance and evaluating the roles of and interactions between genetic variants, which influence the metabolic, inflammatory, and immune processes in the pathogenesis of cardio-metabolic diseases, may better explain an individual's risk of developing diabetes and its complications. Secondly, we validated the effects of summed risk alleles in an independent Korean population. Finally, combining multifactorial polygenic risk factors may be beneficial for better risk prediction in patients with CVDs.

## Supporting information

**S1 Table. Weighted polygenic risk scores from the genome-wide association study.**
(PDF)

**S2 Table. Application of weighted polygenic risk scores according to subtypes of acute ischemic stroke (AIS).**
(PDF)

## Acknowledgments

The authors wish to thank all of participants, patients and staffs for manufacturing "The First Korean Stroke Genetics Association Research" study and providing technical support for clinical validation research.

## Author Contributions

**Conceptualization:** Eun Pyo Hong, Jun Hyong Ahn, Jin Pyeong Jeon.

**Data curation:** Dong Hyuk Youn, Bong Jun Kim, Jae Jun Lee, Doyoung Na.

**Formal analysis:** Eun Pyo Hong, Bong Jun Kim.

**Funding acquisition:** Jin Pyeong Jeon.

**Investigation:** Eun Pyo Hong, Jun Hyong Ahn, Jin Pyeong Jeon.

**Methodology:** Eun Pyo Hong.

**Project administration:** Jin Pyeong Jeon.

**Supervision:** Jin Pyeong Jeon.

**Visualization:** Eun Pyo Hong.

**Writing – original draft:** Eun Pyo Hong, Bong Jun Kim, Jun Hyong Ahn, Jeong Jin Park, Jong Kook Rhim, Heung Cheol Kim, Hong Jun Jeon, Gyojun Hwang, Jin Pyeong Jeon.

**Writing – review & editing:** Eun Pyo Hong, Dong Hyuk Youn, Bong Jun Kim, Jae Jun Lee, Doyoung Na, Jun Hyong Ahn, Jeong Jin Park, Jong Kook Rhim, Heung Cheol Kim, Hong Jun Jeon, Gyojun Hwang, Jin Pyeong Jeon.

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
