## [Decision Letter · Decision Letter 0]

26 Dec 2021

PONE-D-21-32285Genome-wide polygenic risk impact on the intracranial aneurysm and acute ischemic strokePLOS ONE

Dear Dr. Pyeong Jeon,

Thank you for submitting your manuscript to PLOS ONE. After careful consideration, we feel that it has merit but does not fully meet PLOS ONE’s publication criteria as it currently stands. Therefore, we invite you to submit a revised version of the manuscript that addresses the points raised during the review process.

Please carefully evaluate what was suggested by the reviewers, especially in terms of size of the sample studied and then provide the necessary elements (statistical and otherwise) to overcome this criticality.

We look forward to receiving your revised manuscript.

Kind regards,

Giuseppe Novelli

Academic Editor

PLOS ONE

Journal Requirements:

3. We note that you have stated that you will provide repository information for your data at acceptance. Should your manuscript be accepted for publication, we will hold it until you provide the relevant accession numbers or DOIs necessary to access your data. If you wish to make changes to your Data Availability statement, please describe these changes in your cover letter and we will update your Data Availability statement to reflect the information you provide

Reviewers' comments:

Reviewer's Responses to Questions

**Comments to the Author**

1. Is the manuscript technically sound, and do the data support the conclusions?

Reviewer #1: Yes

Reviewer #2: Yes

2. Has the statistical analysis been performed appropriately and rigorously? 

Reviewer #1: Yes

Reviewer #2: N/A

3. Have the authors made all data underlying the findings in their manuscript fully available?

Reviewer #1: Yes

Reviewer #2: No

4. Is the manuscript presented in an intelligible fashion and written in standard English?

Reviewer #1: Yes

Reviewer #2: Yes

5. Review Comments to the Author

Reviewer #1: In this manuscript the authors aim to shed light on hypothetical shared genetic risk factors for Intracranial Aneurysm (IA) and Acute Ischemic Stroke (AIS), presenting and analysing risk models based on weighted Polygenic Risk Score (wPRS) derived from Genome-Wide Association Studies (GWAS) previously published by the same authors.

For what is in my competence, the experimental design sounds carefully conceived and the manuscript is well written. Statistics are described in details and conclusions are reported in a clear and appropriate fashion. However, as the same authors stated in the Discussion section, I must observe that the study has limitations, i.e. small sample size and lack of an investigation on associations between AIS subtypes and IA, which could be of major interest in order to implement the predictive power of the analysis. This study, on his current form, provides a promising starting point and it can have an impact in terms of designing broader and deeper investigations.

On the other hand, I think that the discussion addresses relevant topics, such as the importance of the introduction of PRS derived models to improve risk stratification and the lack of generalizability to non-European ancestry population of existing models. A larger investment on the collection of studies from non-European ancestry is definitely needed.

Reviewer #2: The Authors claim to determine whether the polygenic risk score developed from intracranial aneurysm patients has a common genetic basis with acute ischemic stroke in a Korean population. To do that they applied a weighted PRS model based on a previous genomewide GWAS study using 250 intracranial aneurysm patients in hospital-based multicenter cohort and a validation study in 222 patients who suffered acute ischemic stroke. The work of the Authors is interesting and points out a different way to approach these conditions, which are known to share several clinical risk factors. However, to the date, due to the small size of the sample and the lack of identification of acute ischemic stroke subtypes the use of PRS in common clinical practice isn't feasible yet. Some typos should be revised. Neverthless Authors' work is worthy of further investigation on a larger cohort which a more accurate clinical subtyping.

6. PLOS authors have the option to publish the peer review history of their article (what does this mean?). If published, this will include your full peer review and any attached files.

Reviewer #1: No

Reviewer #2: No

---

## [Author Response · Author response to Decision Letter 0]

21 Jan 2022

Manuscript ID: PONE-D-21-32285

“Genome-wide polygenic risk impact on the intracranial aneurysm and acute ischemic stroke” 

We are submitting a revised version of the above manuscript according to the letter from the Editorial Committee. We have made corrections and clarifications in the manuscript based on reviewers’ comments. In this revision manuscript, we inserted two additional authorships (JJL and DN) because those coauthors contributed to data curation of stroke subtypes and writing to the review process. We highlighted the revised text in gray color in the revised manuscript (filename: Revised Manuscript.docx), the revised manuscript without gray highlighted (filename: Manuscript.docx), and attached the supplementary material (filename: Online_Supplemental_Data.pdf) and the updated figures corrected by PACE (https://pacev2.apexcovantage.com/) (i.e., filenames: Fig1 to Fig4.tif).

We described all responses to Reviewer's comments in the file "Revised letter to Reviewer Comments (PDF)" attached by PLOS ONE upload system as well as informed by here (See PDF including detail information such as Supplementary Tables).

Comments from the reviewer 1

Comment 1: In this manuscript the authors aim to shed light on hypothetical shared genetic risk factors for Intracranial Aneurysm (IA) and Acute Ischemic Stroke (AIS), presenting and analysing risk models based on weighted Polygenic Risk Score (wPRS) derived from Genome-Wide Association Studies (GWAS) previously published by the same authors. For what is in my competence, the experimental design sounds carefully conceived and the manuscript is well written. Statistics are described in details and conclusions are reported in a clear and appropriate fashion. However, as the same authors stated in the Discussion section, I must observe that the study has limitations, i.e. small sample size and lack of an investigation on associations between AIS subtypes and IA, which could be of major interest in order to implement the predictive power of the analysis. This study, on his current form, provides a promising starting point and it can have an impact in terms of designing broader and deeper investigations.

On the other hand, I think that the discussion addresses relevant topics, such as the importance of the introduction of PRS derived models to improve risk stratification and the lack of generalizability to non-European ancestry population of existing models. A larger investment on the collection of studies from non-European ancestry is definitely needed.

Answer: Thank you very much for the positive comments on our study. However, as you mentioned, the main limitations are that the sample size in this study was underpowered and the wPRS assessments according to the ischemic stroke subtype based on a large number of patients is an ongoing project. Due to the nature of bioinformatics, if the sample size of the main data in a GWAS is insufficient, the study outcome will be underpowered. Realistically, a way to address this issue is to reduce the false positives associated with diseases by adding a fine-mapping analysis. However, fine-mapping analysis is an alternative approach to discover candidate variants associated with complex traits based on GWAS summary statistics (Reference 1 below). Thus, the best solution would be to increase the number of patients with IA in a future study. Regarding the second limitation, we performed a subsequent analysis of wPRS assessments in the subtypes of acute ischemic stroke (AIS) including cardioembolism (CE, n= 50), large artery atherosclerosis (LAA, n = 72), small-vessel occlusion (SVO, n = 75), and undetermined (UD, n = 25) (Supplemental Table S2 below). Overall, IA-predicting wPRSs increased the risk of all subtypes of AIS with predictabilities between 0.794 and 0.836. Nevertheless, the sample size in each type of stroke was also small. Thus, additional analysis of a large number of AIS patients is necessary. We included these limitations in the Discussion section (page 10 and lines 203-214).

Reference 

1. Schaid DJ, Chen W, Larson NB. From genome-wide associations to candidate causal variants by statistical fine-mapping. Nat Rev Genet. 2018;19(8):491-504. http://doi.org/10.1038/s41576-018-0016-z PMID: 2984461 

S2 Table. Application of weighted polygenic risk scores according to subtypes of acute ischemic stroke (AIS) (See detail in PDF)

Comments from the reviewer 2

Comment 1: The Authors claim to determine whether the polygenic risk score developed from intracranial aneurysm patients has a common genetic basis with acute ischemic stroke in a Korean population. To do that they applied a weighted PRS model based on a previous genome wide GWAS study using 250 intracranial aneurysm patients in hospital-based multicenter cohort and a validation study in 222 patients who suffered acute ischemic stroke. The work of the Authors is interesting and points out a different way to approach these conditions, which are known to share several clinical risk factors. However, to the date, due to the small size of the sample and the lack of identification of acute ischemic stroke subtypes the use of PRS in common clinical practice isn't feasible yet. Some typos should be revised. Nevertheless, Authors' work is worthy of further investigation on a larger cohort which a more accurate clinical subtyping.

Answer: Per your comments, the relatively small sample size in the study and the absence of an analysis of the acute ischemic stroke (AIS) subtypes by PRS were acknowledged as the main limitations of the study. Due to the nature of bioinformatics, if the sample size of the main data in a GWAS is insufficient, the study outcomes will be underpowered. Realistically, a way to address this issue is to reduce the false positives associated with diseases by adding a fine-mapping analysis. Nevertheless, fine-mapping analysis is an alternative approach to discover candidate variants associated with complex traits based on GWAS summary statistics (Reference 1 below). Thus, the best solution would be to increase the number of patients with IA in a future study. Regarding the second limitation, we performed a subsequent analysis further of wPRS assessments in the subtypes of acute ischemic stroke (AIS) including cardioembolism (CE, n= 50), large artery atherosclerosis (LAA, n = 72), small-vessel occlusion (SVO, n = 75), and undetermined (UD, n = 25) (Supplemental Table S2, below). Overall, IA-predicting wPRSs increased the risk of all subtypes of AIS with predictabilities between 0.794 and 0.836. Nevertheless, the sample size in each stroke subtypes was also small. Thus, additional analysis of a larger number of AIS patients is necessary. We included these limitations in the Discussion section (page 10 and lines 203-214).

Furthermore, English grammar and typographical errors were checked again by English proofreading performed by the native speakers and scientific expertise.

.

Reference 

1. Schaid DJ, Chen W, Larson NB. From genome-wide associations to candidate causal variants by statistical fine-mapping. Nat Rev Genet. 2018;19(8):491-504. http://doi.org/10.1038/s41576-018-0016-z PMID: 2984461

---

## [Editor Report · Decision Letter 1]

7 Mar 2022

Genome-wide polygenic risk impact on intracranial aneurysms and acute ischemic stroke

PONE-D-21-32285R1

Dear Dr.Jin Pyeong Jeon,

We’re pleased to inform you that your manuscript has been judged scientifically suitable for publication and will be formally accepted for publication once it meets all outstanding technical requirements.

Kind regards,

Giuseppe Novelli

Academic Editor

PLOS ONE
---

## [Editor Report · Acceptance letter]

30 Mar 2022

PONE-D-21-32285R1 

Genome-wide polygenic risk impact on intracranial aneurysms and acute ischemic stroke 

Dear Dr. Jeon:

I'm pleased to inform you that your manuscript has been deemed suitable for publication in PLOS ONE. Congratulations! Your manuscript is now with our production department. 

Kind regards, 

on behalf of

Prof. Giuseppe Novelli 

Academic Editor

PLOS ONE